# Next Generation Computing and Communication Hub for First Responders in Smart Cities

**DOI:** 10.3390/s24072366

**Published:** 2024-04-08

**Authors:** Olha Shaposhnyk, Kenneth Lai, Gregor Wolbring, Vlad Shmerko, Svetlana Yanushkevich

**Affiliations:** 1Biometric Technologies Laboratory, Schulich School of Engineering, University of Calgary, Calgary, AB T2N 1N4, Canada; kelai@ucalgary.ca (K.L.); vshmerko@ucalgary.ca (V.S.); 2Cummings School of Medicine, University of Calgary, Calgary, AB T2N 4N1, Canada; gwolbrin@ucalgary.ca

**Keywords:** first responders, smart city, computing and communication hub, machine learning and reasoning, persons with disabilities, emergency management cycle

## Abstract

This paper contributes to the development of a Next Generation First Responder (NGFR) communication platform with the key goal of embedding it into a smart city technology infrastructure. The framework of this approach is a concept known as SmartHub, developed by the US Department of Homeland Security. The proposed embedding methodology complies with the standard categories and indicators of smart city performance. This paper offers two practice-centered extensions of the NGFR hub, which are also the main results: first, a cognitive workload monitoring of first responders as a basis for their performance assessment, monitoring, and improvement; and second, a highly sensitive problem of human society, the emergency assistance tools for individuals with disabilities. Both extensions explore various technological-societal dimensions of smart cities, including interoperability, standardization, and accessibility to assistive technologies for people with disabilities. Regarding cognitive workload monitoring, the core result is a novel AI formalism, an ensemble of machine learning processes aggregated using machine reasoning. This ensemble enables predictive situation assessment and self-aware computing, which is the basis of the digital twin concept. We experimentally demonstrate a specific component of a digital twin of an NGFR, a near-real-time monitoring of the NGFR cognitive workload. Regarding our second result, a problem of emergency assistance for individuals with disabilities that originated as accessibility to assistive technologies to promote disability inclusion, we provide the NGFR specification focusing on interactions based on AI formalism and using a unified hub platform. This paper also discusses a technology roadmap using the notion of the Emergency Management Cycle (EMC), a commonly accepted doctrine for managing disasters through the steps of mitigation, preparedness, response, and recovery. It positions the NGFR hub as a benchmark of the smart city emergency service.

## 1. Introduction

‘First responders’ is an umbrella term for the core personnel that provides safety and security for people [1]. This personnel includes firefighters, police, rescuers, paramedics, and emergency medics. Their mission is to respond to various emergencies and disasters and protect the safety, property, and lives of people affected by those events. Cities and states provide first responders with a wide spectrum of operational and strategic resources, including information about infrastructure and current state monitoring data [2]. These resources must be exploited in a rational and economical manner. City services and resources are becoming increasingly digitized, transparent, and ‘smart’; their dynamics can be analyzed and predicted using Artificial Intelligence (AI)-enabled technologies [3,4,5,6]. This leads to the problem of embedding first responder technologies into smart city technology infrastructure. A decade ago, technological advances in information and communication prompted the rise of the Next Generation First Responder (NGFR) technologies [7]. Conceptually, the NGFR hub is a practice-centric technology extrapolation solution for emergency service.

Since then, many advanced studies such as [8,9,10,11,12] created a technological framework for the NGFR. Nowadays, it is a standard R&D practice to explore a wide spectrum of tools, from cloud and Internet of Things (IoT) resources [2,4,13], to devices for detecting humans and animals behind the walls [14,15,16].

Development of the NGFR AI-enabled hub is a logical phase in systematizing previous works [7,17]. Integrating advanced technologies in a practice-centric manner and open design of on-body devices allows for improvement and innovation, using interoperability and compatibility standards and protocols. To carry out these and other emergency and rescue tasks, an on-body computation and communication network must be created. A central part of the on-body network includes AI-based tools for the responders’ Cognitive Workload (CW) monitoring and prediction [18,19,20,21], health and emotional state [9,22,23], and stress control and prediction [24,25].

In this paper, two emerging architectures serve as the building bricks for the NGFR hub:An NGFR SmartHub on-body system developed by the US Department of Homeland Security (DHS) [7];WiLIFE public safety platform developed by the University of Florence, Italy, and partners [8].

Using such a platform, we take the next step and show how to embed it into a smart city infrastructure. For this, standard categories and indicators for smart city evaluation will be employed [26,27]. The NGFR hub shall also bridge the conceptual gaps between centralized and distributed computing, as well as exploit the cloud-deployed resources. The NGFR can also be viewed as practice-centric technology roadmapping. The NGFR hub’s expandable architecture will be able to integrate new devices and systems as they become available. To address the systemic questions of the NGFR hub’s robustness, we consider the employed technology roadmapping within the coordinates of the Emergency Management Cycle (EMC) [28,29]. EMC assumes the four states of mitigation, preparedness, response, and recovery. This is a useful generalization of the practice-centric view [7]. Therefore, we assert that the NGFR hub, as an integrated part of a smart city, must be designed using the EMC doctrine. The above motivates the strategic goal of this paper, a synchronization of the R&D of the NGFR hub with the smart city’s resilience to crises and disasters.

The aforementioned problem representation and goal formulation also raise numerous research questions regarding the new technological relationships between the NGFR hub and smart city infrastructure. In particular, nowadays the requirement for state-of-the-art technologies is that they be, by design, inclusive to people with disabilities [30,31,32,33,34,35], and smart cities have mechanisms, such as IoT and distributed ledger, to provide disabled citizens with tools for using e-health, and social services [13]. The NGFR will have access to the related records of those individuals throughout their mission.

In this paper, we consider a case study of how the NGFR hub can provide resources for NGFR to assist individuals with disabilities better. This additional task requires particular skills from first responders to recognize individuals with disabilities, identify the type of impairment, and interact with these individuals in order to provide help. It also includes technological support that can be provided by specially trained AI-enabled assistants [36,37]. This is an open problem and a challenge [38,39].

In summary, the key motivation of this work is to examine and extend the concept of SmartHub, a breakthrough practice-centric solution to support the NGFR. The NGFR must be enabled to execute multiple tasks under incomplete and uncertain information, unreliable equipment, and external support in the adversarial environment. We outline the R&D roadmapping and provide strategic direction to consolidate the R&D efforts. Our main goal is to lay out the ways to embed the extended SmartHub into the smart city. We outline how to satisfy the requirements for such embedding in terms of standard categories and indicators of smart city performance. We focus on conditions of critical importance, such as interoperability and standardization, the potential for extensions, modeling, and improving NGFR technologies to assist citizens with disabilities. Our extensions of the SmartHub address two complex tasks: CW monitoring and emergency assistance for people with disabilities. In solving complex problems, the aggregation of different methods often provides reliable assessments of events and processes involved in such. Following this prerogative, we developed an AI formalism—an ensemble of machine learning processes aggregated using machine reasoning for both tasks. These multiple targets prompt the strategic goal of our study—synchronization of the R&D towards the smart city’s resilience to crises and disasters. This strategic goal leads to the commonly accepted EMC doctrine that is innovatively interpreted in our work in terms of the R&D technology gaps. The quintessence of our approach is the NGFR hub, the modified version of the SmartHub.

The rest of this paper is organized as follows. In Section 2, the related works are reviewed. The problem is formulated in Section 3, and the NGFR hub architecture is introduced in Section 4. The results of embedding the NGFR hub into the smart city infrastructure are reported in Section 5, and a case study with experimental sondage is provided in Section 6. We discuss how to support the NGFR in operations with people with disabilities in Section 7. Open problems and potential solutions are discussed in Section 8, and key conclusions are provided in Section 9.

## 2. Related Work and R&D Landscape

First responder technologies are an integrated part of a smart city’s emergency management infrastructure [1,8,27,40,41]. The report by the National Institute of Standards and Technology (NIST) [11] reviewed the technologies used by first responders prior to 2019. It revealed that there is a need for devices that provide information about and for first responders in real time. Another report by the U.S. Pacific Northwest National Laboratory and the First Responder Technology Alliance of the U.S. Department of Homeland Security (DHS) [9] reviewed the advanced technologies for first responders such as physiological and biological sensors, heads-up and body-worn visual displays, body-worn sensors, smart glasses, self-powering technologies, hands-free ergonomically optimized communication systems, body-worn cameras, and breathing apparatus. Our brief follow-up audit of these and other works identifies the main directions of the R&D in the area of emergency management:Public area networks for first responder needs that are provided by smart city infrastructure, e.g., [6,8,42], including big data stream [2,13,43] and unmanned aerial surveillance [18,44];Intelligent tools and equipment, such as (a) through-the-wall detectors for enhancing search and rescue missions under limited visibility conditions caused by smoke, walls, and collapsed debris [16]; these detectors can be placed on autonomous unmanned aerial vehicles [14]; (b) robotic systems for vital sign determination [15,45,46]; this might be useful when the assessments of key triage vital signs (respiratory rate, pulse rate, and capillary refill time) by medical professionals may be unavailable or inaccurate given extreme circumstances;Predictive analytics to assist with pedestrian emergency evacuation [47], crime feature detection [48,49], fire prediction [50], and rockfall and landslide monitoring [51];On-body network of sensors, computing, and communication tools for first responders, e.g., [21,23,24].

A critical trend in the above directions is the redistribution of computing and communication resources: the tasks of on-body mobile computing tools are delegated to the city infrastructure and cloud platform, leaving only energy-saving pre-processing computing. The framework for such relocation includes breakthrough technological innovations such as distributed ledger [41,52] and IoT [2,8,13] based on cloud resources. Thanks to these new opportunities, on-body networks become the communication hubs explored by the DHS’ SmartHub on-body architecture [7].

To the best of our knowledge, the most relevant architecture to the SmartHub is the WiLIFE emergency management platform [8] that provides a technological avenue to the emergency infrastructure of a smart city focusing on deployment. The core concept of the SmartHub architecture is creating a communication hub for on-body resources (e.g., sensing networks and preliminary processing) and external cloud-hosted resources available via a distributed ledger and IoT, as well as other smart city resources. This taxonomical view is useful for R&D planning, technology roadmapping, and the aggregation of various technologies and computing paradigms.

## 3. Problem Formulation and Approach

In our study, a smart city is defined as an urban area with a highly developed sustainable infrastructure and advanced communication systems. The NGFR platform, which represents a computing and communication hub, must satisfy both the general emergency response requirements (e.g., emergency detection, EMC, hazards taxonomy, and crowdsourcing) [6,42,51,53], and the technological-societal requirements. The latter addresses interoperability with the smart city emergency resources, including surveillance, IoT, positioning systems, mobile radio, and various predictive technologies regarding transportation, crime, epidemic, and fire [50,54,55,56]. The first response quality critically depends on the societal-technology resources delegated by the city. Taking these requirements into account, the research question is **how to embed the NGFR platform into smart city technology infrastructure?** To solve this problem, we propose the following task-oriented approach:

**The first task** is to choose the NGFR platform from the identified candidates; it requires an audit of the existing body of work. The priority of such an audit is to focus on the practice-centric distributed architectures. The previous section partially contributes to the exploration of the R&D landscape.

**The second task** involves examining the NGFR platform with respect to the standard categories and indicators [26,27]. This means defining the conditions for embedding the NGFR hub into the smart city infrastructure. Although these conditions provide a general picture, the details are clarified by the case studies.

**The third task** is to extend the NGFR platform, focusing on the improvement of the NGFR performance using a commonly accepted approach to the CW monitoring and on the facilitation of emergency services for people with disabilities. This task requires originality of approach, and thus, we propose to deploy the advanced methodology known as self-aware computing, which is also a framework for the digital twin. Another extension concerns the accessibility of the NGFR hub, specifically for people with disabilities. In particular, emergency response scenarios require the NGFR hub to include AI technologies for interaction in emergency scenarios, e.g., recognizing persons with disabilities, as well as identifying the type of disability (hearing, visual, or mental). In this paper, we chose an experimental sondage as a tool for a feasibility study of the proposed approach.

**The final task** is a follow-up assessment step to demonstrate how the NGFR hub can help bridge the relevant technological-societal gaps. This includes, for example, a trade-off between the mission priorities and the availability of the resources. Technological-societal roadmapping is a R&D demand.

## 4. NGFR Platform: Preparedness for Embedding

An NGFR platform fits the concept of a smart city. However, many technological-societal problems must be solved in order to embed the NGFR platform into the smart city’s technology infrastructure. This section explains how to create a composite NGFR platform, as well as provides definitions and specifications of structural and architectural components.

### 4.1. Composite NGFR Platform

There are various sources to create a composite NGFR platform. Among first responder platforms such as [6,7,8,18,42], we narrowed down these two: (1) WiLIFE computing platform [8] and (2) NGFR SmartHub platform [7]. Both platforms can be defined as further-generation computing architectures for emergency response in a smart city, both extensively use distributed resources.

In our study, we adopted and extended the SmartHub platform [7] combined with updated specifications of WiLIFE [8] and advances in responder technology components [1,9,10,12,21,24,57].

We call the resulting platform an **NGFR hub**. It is a computing platform based on the distributed computing paradigm using on-body and external resources. We focus our investigation on the embedding of the NGFR hub into the smart city infrastructure. An important focus in this hub is the tools to access the CW of first responders, as well as tools to interact with people with disabilities during emergencies (e.g., identify their language, type of disability, and resources needed). We conduct an experimental sondage of the first responders’ CW assessment assuming the usage of an on-body wireless sensing network.

**Definition** **1.** ***Next Generation First Responder (NGFR)*** *is defined as a group of emergency personnel who are [7] “(1) Protected by next-generation multi-hazard personal protective equipment incorporating technology advances in thermal, ballistic, stab penetration, and chemical and biological agent protection; (2) Connected through secure, integrated, and resilient voice and data communications technology; and (3) Fully aware, thanks to an integrated body-worn system of sensors and enhanced situational awareness and decision support devices”.*

### 4.2. Specification of the Architectural Framework

**Definition** **2.** ***The NGFR SmartHub architecture*** *is defined as an open computing and communication hub for distributed on-body resources (e.g., a wearable network of wireless sensors, preliminary processing of physiological signals, environmental cameras, and portable devices) and external resources (e.g., cloud-based computing) [17].*

Figure 1 illustrates how the NGFR on-body SmartHub architecture (down plane) is embedded into smart city technology infrastructure (upper plane) using Input/Output (I/O) devices (heads-up displays, wrist-worn displays, microphone/earphone headsets, hand-held touchscreen displays, voice-activated commands, etc.). SmartHub data shall be encrypted using AES-256 encryption when stored on-body and when sent off-body. Encryption of data ensures that the data read by unauthorized users retain a level of security by obfuscating the data.

The NGFR SmartHub contains a Controller Module, Power Module, Communication Module, Sensor modules, and Operator Input/Output (I/O) Devices. The Controller Module has internal capabilities such as communications (e.g., Bluetooth, Wi-Fi, USB), audio/video recording, and data storage. Various applications should be available, such as messaging, physiological sensor system management, alerting system management, and voice-to-text for messaging and application commands. The Communications Module provides an interface between the Controller Module and external communications devices. The Power Module provides long-term, exchangeable, and rechargeable battery power to the various modules for extended use. Sensor Modules consist of physiological sensors, chemical, biological, radiological, nuclear, and explosive sensors, thermal sensors, physical sensors, kinesthetic, vestibular, and haptic, etc. The modules communicate with the Controller Module via wired or wireless connections. Each sensor would have its own short-term power source and built-in intelligence with the capability to communicate sensor identification and sensor data to the Controller Module. Sensors could be body-worn (e.g., body cameras, radiation sensors, physiological sensors, etc.) or hand carried. Operator I/O Devices include heads-up displays, wrist-worn displays, microphone/earphone headsets, hand-held touchscreen displays, voice-activated commands, etc. Table 1 provides details and abbreviations used in Figure 1.

Within the NGFR hub, we suggest that common ground for intelligent communication networks shall be a concept of intelligent or cognitive, radio. Cognitive radio is defined as a radio that can sense and understand the radio environment and policies and monitor usage patterns and users’ needs; autonomously and dynamically adapts according to the radio environment, so that it can achieve predefined objectives, such as efficient utilization of spectrum; and learn from the environment and the results of its own actions, so it can further improve its performance Intelligent radio is considered a key enabler for deploying technologies that require high connectivity, such as Wide Area Network (WAN), Public Safety Land Mobile Radio (LMR), Public-Safety Access Point (PSAP), and IoT, including Internet of Flying Things. The cognitive radio with the AI-enabled spectrum sensing techniques became a core of contemporary intelligent communications [58].

Table 1 also represents the NGFR hub extensions related to

−E-health, a multidimensional sustainability system that includes technology, organization, economic, social, and resource assessments. According to the World Health Organization (WHO) [59], e-health encompasses “multiple interventions, including telehealth, telemedicine, mobile health, electronic medical or health records, big data, wearables, and even artificial intelligence”. E-health is an integrated part of smart city and NGFR health monitoring in pre-mission (e.g., training), mission (usage of personal data from e-health databases), and post-mission (e.g., health control, post-traumatic state monitoring).−NGFR CW, the level of measurable cognitive effort needed by NGFR in response to one or more cognitive tasks; the data are transmitted to a commanding operator or a mission control center.−Data from assistive devices focusing on Interacting with People with Disabilities (IPD) [60].−Next Generation (NG) of PSAP such as projects [10,12], that will be able to provide sensor health data to the first responders as well as data from assistive devices, e.g., speech synthesizers and symbol dashboards. Using the NG PSAP channel, new functions of the NGFR platform can be integrated. Note that contemporary PSAP supports only voice calls.

**Table 1 sensors-24-02366-t001:** Basic communication channels of the on-body NGFR hub architecture and its extensions.

Channel	Description
**Framework of the NGFR hub**	
Incident Area Network (IAN)	The network provides a reliable and lively routing path during disaster recovery and emergency response operations when infrastructure-based communications and power resources have been destroyed and no routes are available.
Wide Area Network (WAN)	A connected collection of telecommunication networks distributed across a large geographic area spanning multiple cities, territories, or nations.
Global Positioning System (GPS)	Receive dispatch information containing the incident location in text form, which is information for the responder’s location of the event on the responder’s satellite-based radio navigation system GPS display.
Public Safety Access Point (PSAP)	Receive messages from radio calls, computer-aided dispatch, and other information from PSAP or incident commander containing the location, data, descriptions, and other information regarding the emergency situation.
Land Mobile Radio (LMR)	A person-to-person voice communication system consists of two-way radio transceivers that can be stationary, mobile, or portable.
Media Access Control (MAC)	A network standard that controls the hardware responsible for interaction with the transmission medium.
**Extensions of the NGFR hub (this paper)**	
E-health service	E-health is a cost-effective and secure use of information and communication technologies to support health and health-related fields [59]. It is considered in the context of IoT [13,61].
Cognitive Workload (CW)	Wearable sensors and bio-patches provide the data. Preliminary processing is performed using on-body tools, and the data are transmitted to the cloud and processed using deep learning and machine reasoning for CW assessment.
Interacting with People with Disabilities (IPD)	In the NGFR hub, the IPD addresses a guiding intelligent tool for interaction in an emergency situation with people with different impairments, e.g., visual, hearing, speech, cognitive, and physical impairment (see details and open problems in Section 8).

## 5. Embedding the NGFR Hub into the Smart City’s Technology Infrastructure

This section introduces a rigorous approach to embedding the NGFR platform into the technology infrastructure of a smart city. The embedding method is based on the smart city standard categories and indicators.

### 5.1. Categories and Indicators

We will measure the technology’s performance using various indicators to assess and track the progress of a city or a community on sustainability, smartness, competitiveness, quality of life, etc. [62,63,64]. The measurable categories are constructed from indicators for tracking more holistic progress. For example, the category of smart city emergency services consists of a set of indicators such as response time, the number of emergency stations, equipment, and the cost of response. Hence, the performance of the NGFR hub is a part of smart city performance measurement with task-specific extensions called sub-indicators: $$\mathrm{Smart}\;\mathrm{City}\to \underset{\mathrm{Performance}\;\mathrm{measures}\;\mathrm{and}\;\mathrm{embedding}\;\mathrm{conditions}}{\underset{︸}{\mathrm{Categories}\to \mathrm{Indicators}\to \mathrm{Sub}-\mathrm{Indicators}}}$$

In this scheme, the performance measures are also the embedding conditions.

The smart city categories and indicators are defined as follows.

**Definition** **3.** ***An indicator*** *is defined as a measurable domain-specific characteristic of smart city infrastructure.*

For example, ISO standard [26] defines 100 indicators across 17 themes (or categories) to measure the city service performance and the citizens’ quality of life.

**Definition** **4.** ***A category*** *is defined as an aggregate combination of indicators according to their correlation, causal relationships, or other criteria.*

Details can be found in [27]. Formal methods for identifying categories and indicators, their assessments, and prediction are developed in [65]. In our approach, the notion of indicators and categories is used to study the NGFR hub as an integral smart city performance measure, as well as the conditions for embedding.

### 5.2. Aggregation of Indicators

The NGFR hub is a hub for the “aggregation” of various computational and communication facilities. Notions of categories and indicators are inherently acceptable for describing the NGFR hub.

**Definition** **5.** ***An “aggregative” or “composite” indicator*** *is defined as the compilation of individual indicators into a single indicator based on an underlying model. Aggregate indicators are generally designed to measure multidimensional concepts that cannot be captured by a single indicator [65].*

The aggregative indicators can measure concepts that are linked to well-known and measurable phenomena (e.g., known categories), as well as novel phenomena called integrated or composite indicators. For example, the international standard ISO 37120 [26] recommends a set of core indicators (*c*) and supporting (*s*) indicators for performance assessment of city services and quality of life. The aggregative NGFR indicators include, directly and indirectly, energy (c=4,s=3), environment (c=3,s=5), fire and emergency response (c=3,s=3), governance (c=2,s=4), health (c=4,s=3), safety (c=2,s=3), shelter (c=1,s=2), Telecommunication and innovation (c=2,s=1), Transportation (c=4,s=5), and water and sanitation (c=4,s=3). In our approach, a measurable phenomenon in terms of a composite indicator addresses the embedding of the NGFR hub into smart city technology infrastructure.

Figure 2 provides details on the construction of the aggregative NGFR indicator. There, the 1st pane contains a sample of standard smart city categories; the 2nd pane, called an aggregative NGFR indicator, contains a sample of essential NGFR indicators; the 3rd pane contains the NGFR sub-indicator (or apps). The indicators of the 2nd pane cover all categories of the 1st pane. This leads to the following conclusions: (1) the NGFR technologies must be provided by city infrastructure for efficiency of emergency and safety service; and (2) the NGFR can be used as a benchmark in digital twin modeling, e.g., [3].

Robustness of the NGFR hub is achieved by the apps that are deployed on a cloud platform; they are encoded as sub-indicators. Various extensions are possible, such as cognitive workload monitoring assistant (Section 6). Any AI-enabled assistant can be connected with the NGFR hub: crime prediction assistant [48,49], stress monitoring assistant [24,25], fatigue monitoring assistant [22], and language assistants [57]. Any innovation can be further improved. For example, hand gesture detectors can be improved by ‘air-writing’ fingertips [66], which is extremely useful in some specific emergency scenarios when voice communication is impossible (e.g., loss of voice as a trauma or related disabilities). The apps for recognition of individuals with disabilities, identification of the type of impairment, and interaction with these individuals in order to provide help are considered in Section 7.

### 5.3. Formal Notions and Framework of Indicators

The formal notions and framework of smart city indicators introduced in report [65] are illustrated in Figure 3 with our update of last decade’s techniques. The first pane of Figure 3 contains names of formal methods and the second one lists the goals of their applications in modeling and describing the indicators. These include traditional machine learning techniques such as factor analysis, principal component analysis, and cluster analysis, and advanced ones such as deep learning networks, as well as powerful AI techniques such as machine reasoning [67], synthetic data generation [68,69], and digital twins [70,71]. Data analytics, including data aggregation, visualization, and correlation analysis, are being extended by contemporary causality analysis and predictive tools. These methods and techniques are used to implement the aggregated NGFR indicator (Figure 2). For example, for CW assessment, we used factor analysis, deep learning networks, and machine reasoning, aiming at the aggregation of reasoning and discovering the correlated and causal relationships of patterns of interest for prediction and risk assessment of critical scenarios.

## 6. Case Study: Digital Twin of the NGFR CW

In this section, we report the results on the NGFR platform extension for CW monitoring. The NGFR hub belongs to an open computing platform that is based on open standards (Definition 2). CW monitoring is a complicated task that requires a lot of computing resources and communication channels, such as the on-body network of wireless physiological sensors and bio patches; access to personal data, such as psychological and geographical data stored in distributed ledgers; and AI-enabled tools for CW monitoring deployed in cloud facilities.

### 6.1. Cognitive Workload

Based on works [72,73,74], the CW assessment of first responders is understood as follows:

**Definition** **6.** ***Cognitive Workload (CW)*** *is defined as the level of measurable cognitive effort needed by an individual in response to one or more cognitive tasks.*

Note that the terms ‘cognitive workload’, ‘cognitive load’, or ‘mental workload’ are used to describe and evaluate the cognitive demand of a task. CW is particularly relevant in emergency and disaster situations. Emergency situations occur suddenly and can change unexpectedly, requiring critical decision-making under personal involvement. This involvement includes physiological health factors, expertise, stress, fatigue, distractions, or other internal psychological states. Therefore, understanding and monitoring CW can provide valuable insights into the performance and well-being of first responders in challenging and dynamic emergency scenarios. This is the key motivation for choosing the CW for the experimental sondage of the NGFR hub.

### 6.2. Digital Twin of the CW

Digital twins have been applied to various problems in smart cities, such as creating 3D representations of smart city entities, incorporating time-series and historical data, heatmaps, geometries, and shapes related to traffic flows, bus routes/stops, and cycling paths [3]. This technology has also found applications in first responder tasks such as global wildfire prediction [50] and fire safety [75]; a guiding system for pedestrian emergency evacuation [47]; and cognitive personalized systems of smart city [76]. Digital twin methodology, including model- and twin-enablingtechniques (e.g., physical-to-virtual and virtual-to-physical techniques), as well as perspectives on the trajectory of digital twin technology over the next decade, can be found in [70,71]. In CW assessment, we refer [77] that introduced the fundamentals of human-in-the-loop digital twin, including cognitive performance (such as workload level, situation awareness, decision-making abilities), emotional state, and behavior. This leads to the definition of human digital twin given in [77] that precisely covers a digital twin of CW:

**Definition** **7.** ***“Human digital twin*** *is defined as a pairing of a real-world human twin and a human digital twin, which includes a model of physical appearance, physiology, personality, perception, cognitive performance, emotion, or ethics of a human; where the real-world human and human digital twin are integrated such that a change in the real-world human or its digital representation produces change(s) in the other” [77].*

Note that this definition is close to the ISO standard definition of digital twin in manufacturing [78]. Definition 7, as well as [70,71,77], implies the three mandatory features of digital twin:−a model of the object (i.e., aggregative learning model);−an evolving set of data relating to the object (i.e., continuous monitoring);−bidirectional communication between the system and its digital representation for dynamically updating or adjusting the model using the continuously monitored data (i.e., mechanism of perception–action cycle).

In this paper, the concept of the human digital twin was applied to the CW of first responders. To assess the CW, the model was composed using a pattern learning sub-model, causal learning sub-model, and deep learning sub-model. An appropriate publicly available dataset was used in experiments. The NGFR hub provides the necessary conditions for bidirectional communication between the first responder and the model.

In the context of the NGFR hub, we should mention the relation of the digital twin and self-aware computing with a human-in-the-loop concept. Our machine learning–reasoning paradigm for the CW is mapped into the computing architecture called self-aware systems [79,80,81]. Applications of the self-awareness principles for the CW tasks are reported, for example, in [19,20]. The current challenge is an extension of the self-aware principle to the team’s CW. The requirements for a digital twin platform in the context of ISO standard [78] are introduced in [82]. Most of them address a self-aware paradigm. It includes, for example, a bidirectional synchronization for the purpose of real-time interactions between real objects and their digital model, as well as requirements for interoperability.

### 6.3. Machine Reasoning Model

Contemporary machine reasoning includes probabilistic causal models and tools rather than traditional rigid logic rules to deduce new information from existing data. In our work, we refer to an approach known as causal reasoning and Bayesian networks [67]. This approach assumes reason not only about statistical associations but also about cause-and-effect relationships. It involves representing and manipulating causal relationships between variables. In particular, we use Bayesian networks, a graphical model that represents causal relationships between variables using conditional probabilities and allows us to infer the likelihood of different outcomes or scenarios given observed evidence.

Using the causal graph models, we derived a computational scheme for the first responder’s CW assessment shown in Figure 4. Three kinds of data are distinguished: (1) demographic data (stored outside the NGFR hub, e.g., in a distributed ledger), (2) psychology or behavior data (partially stored outside the NGFR hub, e.g., in a distributed ledger), and (3) physiological data from wearable sensors deployed in the clothes and body of the first responder. A dataset from [83] was used in an experimental sondage.

Demographic data processing based on pattern learning leads to Reasoning-I. Causal learning is used for processing psychological data, resulting in Reasoning-II. Data from wearable sensors are processed using deep learning and lead to Reasoning-III. These three kinds of machine reasoning are aggregated into a composite Reasoning using a causal mechanism provided by causal networks. In this paper, a Bayesian causal network is chosen.

The computational scheme for CW assessment in Figure 4 is characterized by several useful features:It is commonly accepted that CW is by nature multimodal. However, in practice, research is often limited by physiological data. The scheme in Figure 4 distinguishes three types of multimodal data required for CW assessment: demographic, psychological, and physiological. Moreover, this scheme suggests that such taxonomized data must be learned and processed using adequate computational methods, i.e., exploratory learning, causal learning, and deep learning.This scheme suggests aggregating results of learning at the level of reasoning. For this, various kinds of causal networks can be used, such as Bayesian, fuzzy, Dempster-Shafer and its extension, and credential [84]. The choice depends on the interpretation of uncertainties. In this paper, a Bayesian causal network was chosen.

### 6.4. The CW Mode of NGFR Hub

The CW-monitoring process is described in the previous subsection as well as schematically illustrated by Figure 4 (accept feedback). Figure 5 provides further details and illustrates two structural components:−The NGFR hub’s CW mode as a complete process of the CW monitoring in terms of the perception–action cycle (lower plane);−Embedding the NGFR hub into a smart city infrastructure (upper plane).

These structural components are connected via I/O devices and protocols. The smart city infrastructure provides both cloud-based resources (e.g., machine learning and reasoning) and e-health resources (e.g., personal health information of first responders) to the NGFR hub. Furthermore, the NGFR hub transmits the preliminarily processed data from physiological sensors and bio-patches for further processing on cloud tools.

Consider the operation within the CW mode. The evidence such as the signals from the on-body wearable sensor network, are processedin order to estimate the current health status of a first responder (State Estimator), and inform the human manager (Decision Maker) who can make decisions, e.g., halt the mission. As the mission continues, the CW analyzer learns this data (CW Analyzer); this learning results in creating the CW model composed of both the statistics-based pattern learning and the probabilistic reasoning models such as BN. The BN models the scenario in the form of a joint probability distribution. It is constructed using joint probability distribution derived from pattern recognition as well as other statistical information. The BN is the framework for reasoning that results in risk assessment (Supporter). For details, we refer the reader to a wide spectrum of the literature, including the recent monograph [85]. The assessed risks are reported to the first responder, e.g., “The CW increased by 20% and reached level 3 (medium on the cale of 1–5) in the past time interval (5 min)”. In the next time interval, the health status of the first responder is changed, causing the behavior change in the adversarial environment (flames, falling debris), and the BN model is updated. This computational model of CW is a digital replica of a physical object’s CW status. It continuously evolves in near real time following the behavior of this object [70,71,77].

Note that Figure 5 reflects several advanced computational paradigms for the implementation scheme. First, it is self-aware computing [79,80]. The applications of the self-awareness principle for CW tasks are reported, for example, in [19,20], where a 78% accuracy of the CW detection was achieved. Secondly, this is a human-in-the-loop paradigm.

### 6.5. Experimental Sondage

The goal of the experimental sondage is to justify the feasibility of CW monitoring based on AI formalism given in Figure 4.

#### 6.5.1. Basics of CW Monitoring

CW monitoring is a process that involves at least two multifaceted processes: stress [21,23,24,86] and mental fatigue [22,87,88]. Stress is a complex psycho-physiological pattern, a response of the body to any demand for change. The main feature of stress is that under stress, the primary system vital for survival, i.e., vision, cognitive processing, and motor skills (or all together), can break down. Moreover, stress can be propagated (contagioned) between teammates [89,90]. Mental fatigue is a psycho-physiological state caused by engaging in cognitively demanding activities. The main characteristic of fatigue is the feeling of resistance against further (cognitive) effort.

In practice, the CW, stress, and mental fatigue are assessed using the same psycho-physiological markers, yet with different processing goals. Various databases are publicly available for testing and comparison of processing tools. In the current experimental sondage of the CW, we use the publicly available database CogLoad [83].

#### 6.5.2. Ensemble-Based CW Assessment

Following the processing scheme shown in Figure 4, demographic data from the CogLoad dataset were processed in our experiment using exploratory learning and the simplest pattern learning tools. Psychological data needed more detailed processing, using causal learning. Figure 6 provides some details on the causal relationships between psychological traits using Structural Equation Modeling (SEM) [91]. SEM is a statistical technique used to examine causal relationships among variables. Using statistical software and goodness-of-fit measures, SEM provides a quantitative framework for understanding the complex interconnections within a system.

The nodes of the SEM designed for this case study cover six domains based on the HEXACO survey [92]: honesty–humility domain (e.g., sincerity, fairness, modesty), emotionality domain (e.g., dependence), extraversion domain (e.g., social boldness, sociability, liveliness), agreeableness domain (e.g., gentleness, flexibility, patience), conscientiousness domain (e.g., diligence, prudence), openness to experience domain (e.g., openness). For example, dependence is related to psychological trait by a coefficient of 0.310, which means that for a one-unit increase in dependence, we expect psychological trait to increase by 0.310 units. The positive sign of the coefficient suggests a positive relationship: higher dependence is associated with higher (more Stressed) psychological trait. A negative coefficient indicates an inverse relationship, where an increase in the independent variable is linked to a decrease in the dependent variable.

Let us consider Reasoning-I, noted in Figure 4, which involves utilizing demographic data, age, and sex to predict personality traits and their effects on the CW (Figure 6a). The SEM produced a regression coefficient of −0.011 for variable ‘age’, and −0.145 for variable ‘sex’. This indicates a higher impact of the individual’s sex on personality traits and the individual’s response to workload. When combined, these demographic data strongly impact the CW, as indicated by the regression coefficient of −0.938.

Now, consider Reasoning-II, noted in Figure 4 that focuses on psychological traits as a cumulative of the personality traits illustrated by the SEM fragment in (Figure 6b). The trait ‘Agreeableness’ exhibits a regression coefficient of −0.250, whereas ‘Openness’ has a coefficient of −0.025. The nodes ‘Conscientiousness’ and ‘Honesty-Humility’ have regression coefficients of −0.298 and −0.263, respectively, whereas ‘Dependence’ and ‘Extraversion’ demonstrate positive regression coefficients of 0.310 and 0.156, respectively. This indicates that the prevalence of traits such as Dependence and Extraversion would positively impact an individual’s response to the CW demands.

The created SEM becomes the basis for designing the causal network, a directional acyclic graph representing cause-and-effect relationships between variables. It allows for evaluating the scenarios of interest and performing inference using the variables’ probability distributions, represented by conditional probabilities. This causal network is called a Bayesian causal network (BN). It will be used below for explaining the ensemble machine reasoning scheme shown in Figure 4.

#### 6.5.3. Ensemble Machine Reasoning

A fragment of the BN for prediction of CW based on physiological, demographic, and psychological data from the CogLoad dataset are given in Figure 7. The BN serves as a key component in our ensemble machine learning–reasoning framework. The BN comprises nodes representing critical factors in the CW assessment process: demographics, personality trait, physiological markers, and workload. Demographic and psychological data are combined in a node ‘Personality trait’, which assumes three values: Stressed (S), indicating a state characterized by the cognitive load; Restful (R), signifying a state free from such load; and Neutral (N).

The physiological or biometric measurements were collected by the creators of the CogLoad dataset using a Microsoft Band 2 wristband. These measurements encompass Galvanic Skin Response (GSR), Heart Rate (HR), RR Intervals (RR), and Skin Temperature (Temp), all sampled at a frequency of 1 Hz.

The physiological measurements were processed in our experiment using a machine learning technique such as a neural network, which identifies the measurements that are the best predictors of the CW level, divided into two categories: Rest and Load. The physiological signals were normalized to the range of 0 to 1, followed by a standardization to achieve a mean of 0 and a deviation of 1. Next, feature extraction and classification were performed using deep learning networks: recurrent neural networks and temporal convolutional networks, similar to the approach described in [24]. The binary values, Rest or Load, correspond to the classification output. The outputs indicate the contribution of each physiological measurement as the CW predictor. In particular, the RR predicts Load with a probability of 86.07% and Rest with a probability of 13.93%. These predictions are combined, resulting in an average prediction of 47.14% for Load and 52.86% for Rest.

#### 6.5.4. CW Prediction

Consider several scenarios of the CW level evaluation using the designed BN in terms of its likelihood (belief) given the prior likelihoods and a current observation of selected variables. Table 2 shows several ‘prediction’ scenarios based on causal connections between variables in the BN. In the first prediction task (How would seeing *X* change my belief in *Y*?), preliminary information is that the first responder is a 30-year-old man. It can be predicted that their personality trait is Neutral (Reasoning-I). In the second prediction task (What if? What if I do *X*?), it is assumed that the first responder exhibits a Stressed trait. It can be predicted that their personality trait is assessed to be Stressed rather than Neutral. In the third prediction task (Why? Was it *X* that caused *Y*? What if I had acted differently?), the CW can be predicted to be moderate based on the measurement from one sensor. However, if the second sensor’s measurements also became available, the prediction mechanism would detect the high level of CW. In the fourth task, prediction based on demography knowledge (Reasoning-I), psychology knowledge (Reasoning-II), and physiological knowledge (Prediction-III) is aggregated, resulting in predicting 91% CW of a given first responder (i.e., 30-year-old man).

## 7. NGFR Hub for People with Disabilities

“Disability is part of being human. Almost everyone will temporarily or permanently experience disability at some point in their life. An estimated 1.3 billion people—about 16% of the global population—currently experience significant disability. This number is increasing due in part to population aging and an increase in the prevalence of noncommunicable diseases” [93]. The World Health Organization (WHO) defines disability as “an umbrella term, covering impairments, activity limitations, and participation restrictions” [94]. The objective of this section is to explore how the NGFR hub can effectively provide emergency assistance for individuals with disabilities.

### 7.1. NGFR Interactions with Individuals with Disabilities

Emergency assistance for individuals with various physical, cognitive, and other disabilities is **cardinally different from general emergency assistance**. In the context of the NGFR, the problem is formulated in terms of **discovering technologies and techniques** that aim to help the NGFR provide service of vital importance to people with disabilities in life-critical situations. An essential list of this inclusive approach includes:**Recognize individuals with disabilities.** The indicators for such recognition can be bracelets with the wearer’s health information, essential equipment and supplies such as wheelchairs, walkers, oxygen tanks, batteries, communication devices (head pointers, alphabet boards, speech synthesizers, etc.), and medication [36]. Available AI-enabled pattern recognition tools must be trained to perform such recognition.**Identify the type of impairment.** The framework for this request is based on the International Classification of Functioning, Disability, and Health (ICF) which provides a taxonomy of disability [95]. The ICF contains over 1400 categories, the units of classification. This is utilized to create a Disabled Person Profile (DPP) that describes the disabilities of an individual that are most relevant in the first responder context [60]. The DPP is used to collect operating instructions (e.g., warnings, procedures, and recommendations) for the first responders. However, identifying the type of impairment in conditions such as fire threats and weather cataclysms such as snow, smog, fog, and rain is difficult and requires special techniques and tools.**Interact with individuals in order to provide help.** Given a type of disability, a protocol and recommendations are devised for interacting with the individual. These interactions must align with the disability type, the etiquette, and the interaction skills specific to the above [37].

The NGFR hub concept suggests that the approach to the interaction protocols must be decomposed into two parts: the first part is delegated to the smart city infrastructure, and the second part is related to the NGFR on-body network. Both address assistive technologies for people with disabilities, specifically:−Regulatory, health, and demographic aspects [32];−Current and emerging technologies [33];−Perspectives, needs, and opportunities [34];−Legal and socio-ethical perspectives [35].

The priority assistive products list developed by WHO [96] includes hearing aids, wheelchairs, communication aids, spectacles, artificial limbs, pill organizers, memory aids, and other essential items for many older people and people with disabilities.

More formally, it givesa visual perceptive image that can be obtained by NGFR using video or special devices such as through-the-wall imaging radar, infrared camera, and application of deblurring techniques (needed due to the presence of smoke, fire, or atmospheric turbulence). The task includes the following:

(1) Recognize an individual as being a person with a disability, that is, belonging to the sub-population *S* known in the region of the NGFR service;

(2) Identify the individual’s impairment type, $D\in \{d_{1},d_{2},\ldots d_{n}\}$, demography features $G\in \{g_{1},g_{2},\ldots g_{k}\}$, and retrieve the corresponding IPD protocol $C\in \{c_{1},c_{2},\ldots c_{m}\}$;

(3) conduct interactions using the IPD protocol $C\in \{p_{1},p_{2},\ldots d_{l}\}$ of an individual from *S*, e.g., language, visual, and/or sound signs according to the protocol.

### 7.2. Requirements of the NGFR Technologies

Consider the extensions of the NGFR SmartHub shown in Figure 1 regarding assistive technologies. The first extension addresses the efficiency of the NGFR performance; it was introduced in the previous section. The second extension addresses emergency services for individuals with disabilities, specifically, the recognition of such individuals, the identification of the type of impairment, and interaction in order to provide help. Technically, first responders need the IPD protocol that is available from the smart city’s e-health resources and/or new-generation PSAP.

The technologies and tools for IPD with respect to the type of impairment are differentiated as follows:−For visual impairment [33,97,98,99,100], they include gesture-to-voice technology, alarm signallers with light/sound/vibration, talking/touching watches, braille displays, GPS locators, ultrasound, and infrared based devices;−For hearing impairment [33], they involve text-reader devices, structured sign language, drawing or writing, augmentative communication devices, talking/touching watches;−For speech impairment [33,101,102,103,104], the examples include voice-input voice-output aids, telepresence robotics, mobile communication jackets, eye-gaze tracking systems, and Morse glasses;−For cognitive impairment [33,105,106], there are GPS locators and telepresence robotics;−For physical impairment [30,33,107], there are technologies such as intelligent wheelchairs, exoskeletons, etc.

This categorization implies that the common feature involved a generation of certain signals of the electromagnetic spectrum. These signals form an electromagnetic “signature” of each assistive device or gadget. For an impaired individual, their specific device for interactions can be identified by its electromagnetic signature. For example, study [108] identified two promising technologies to support people with visual impairment: voice-activated digital assistants and apps for smartphones and personal computing devices (WiFi routers, Bluetooth transmitters). All of them can be detected at a distance. Also, voice-input voice-output devices for people with severe speech impairments [101] generate identifiable signals related to speech recognition, message composition, and speech synthesis.

Hence, the NGFR needs an intelligent scanner that is able to automatically scan the frequency band and detect the frequencies of interest, thus identifying the type of the device and, eventually, the type of disability. This shall enable the NGFR to provide emergency interactions with persons who cannot communicate due to injuries, impairment, or being trapped under debris or walls. Such portable scanners must be deployed on the NGFR body, and the corresponding interaction interface must be stored in the e-health resources or other databases provided by the smart city.

We believe that the R&D on such an intelligent scanner will activate additional efforts on design standards [109]. For example, each device must include a signal transmitter specific to the NGFR mission. Note that the related idea is an intelligent or cognitive radio communication [58].

### 7.3. Integration into the NGFR Hub

The requirements for the NGFR technology for emerging IPD formulated in the previous section are incomplete without considering the personalization features of individuals with disabilities. Thus, the NGFR hub must accommodate the individual’s specifics, which shall be stored in the smart city’s e-health database.

Figure 8a represents a formal notion of the personalized assistant. The computational scheme, which was suggested in the first case study (the CW assessment) shown in Figure 4 in this paper, can be adopted. In particular, the demographic data and the type of impairment of an individual can be processed using machine learning methods, whereas personalization and identifying the related IPD features require causal reasoning techniques. For example, reasoning I–IV is aggregated to provide the NGFR with a recommendation for action.

This computational scheme also provides a prediction mechanism for the NGFR to estimate the required personalized IPD under uncertainty due to a lack of data or difficulty in retrieving them at times of disaster. Specifically, a request by an NGFR to provide support to an individual with a disability is transmitted to the e-health resources, processed, and a personalized assistant responds to the request and suggests an initial or partial IPD protocol (Figure 8b). The NGFR interacts with the individual while the data are processed, and a request activates the corresponding e-health resources and provides recommendations to the first responder on the refined IPD and further course of action.

Note that the development of personalized assistants to support first responders in emergency situations in which disabled people are involved was drastically accelerated in the past decade, e.g., [12,60].

## 8. Discussion and Open Problems

There are several emerging topics for discussion in the context of smart cities beyond reported results: (1) societal-technological challenges of the NGFR hub and (2) the EMC as a driver of technology roadmapping.

### 8.1. Societal-Technological Challenges of the NGFR Hub

First responder organizations are integral components of a smart city. There are two societal-technological aspects, including (a) the **readiness** of a smart city to provide resources requested by the NGFR institutions, and (b) an **ability** of the NGFR to efficiently use the provided resources. Their balance has been paid little attention in the R&D community so far. In this context, the NGFR hub is a timely and rational proposal.

Next, the NGFR hub manifests itself as **a mechanism for evaluation** of a smart city technology infrastructure and its readiness to respond to potential crises. This mechanism is based on previous and recent works, e.g., creation of digital twins [3], the IoT as the main driver of service [4], e-health in crisis [13,61], and challenges of crises [6,42].

Finally, the NGFR hub can become a **novel type of societal-technological benchmarks**, i.e., a specific data structure that aggregates a set of standards for the purpose of comparison of various approaches and systems. Intuitively, the NGFR hub satisfies the key societal-technology requirements to the benchmark in the context of a smart city infrastructure: it is relevant (it is an integrated part of a smart city); reproducible (it is able to consistently produce similar results when the NGFR hub is run with the same test configuration); fair (it can compete on their merits without artificial constraints), verifiable (providing confidence that a NGFR hub result is accurate), and usable (avoiding roadblocks for users to run the benchmark in their test environments) [110]. For example, the proposed in [44] benchmark contains about 100K images covering urban scenarios on intelligent transportation, intelligent surveillance, and drones. Algorithms can be compared with respect to occlusion ratio, truncating ratio, illumination, direction, time period, and weather.

Our work suggests creating sets of benchmarks for the NGFR methods and tools based on the concept of on-body computing and communication hub. This approach includes a CW mode as well as a personalized mode for people with disabilities.

### 8.2. EMC as a Driver of Technological-Societal Roadmapping

The concept of the NGFR hub suggests updating assessments of smart city preparedness to potential threats. First responders are involved in local emergencies (e.g., fire, robbery, shooting, catastrophic accident, flood) and global emergencies (e.g., pandemic, climate change effects). Generalization of both kinds of emergency management addresses the commonly accepted doctrine Emergency Management Cycle (EMC) [28,29]. Accordingly, the EMC, any societal-technological system at a certain point of time, is in one of four states: mitigation, preparedness, response, and recovery. In our approach, the NGFR hub can be embedded into the EMC, which helps formulate and answer the following questions in the context of a smart city:−How can the NGFR hub be used to mitigate potential threats? (mitigation phase);−How should the NGFR hub be integrated into the smart city emergency infrastructure? (preparedness phase);−How should the NGFR hub operate in crisis scenarios? (response phase);−How can the NGFR hub contribute to the post-crisis period? (recovery phase).

A useful property of the EMC is the identification of causal technological gaps and developing technology roadmapping. Figure 9 provides a taxonomical view of the NGFR system aimed at identifying the technological-societal gaps using the mechanisms of the EMC projections.

For example, the response phase poses the question of how to return to a normal state (recovery phase) without the loss of efficiency (performance), in preparation for the next potential epidemics (preparedness phase). In this EMC pathway, we identified two kinds of technological-societal gaps:$$\mathrm{Response}\overset{\mathrm{Gap}\;I}{⟶}\mathrm{Recovery}\overset{\mathrm{Gap}\;\mathrm{II}}{⟶}\mathrm{Preparedness}$$
where Gap I and Gap II contain actions, mechanisms, and strategies for returning to a regular mode (recovery) and from the R&D process toward a future potential pandemic (preparedness), respectively. An efficient response is enabled by the efficiency of the mitigation and preparedness phases. That is, it is determined by how well the lessons from the previous crises were learned.
Mitigation⟶Response⟵Preparedness
R&D on the EMC for NGFR is of extreme demand.

### 8.3. Dominating Trends

There are several **dominating trends** aiming at improving the NGFR performance based on advanced technologies:−**Sensing on-body networks** for monitoring and predicting first responders’ physiological state and behavior. In particular, the most recent approach to the CW assessment includes, apart from the monitoring of biometrics or physiological signals, the personality trait and demographics [72]. In this paper, we took the further step –we developed and experimentally demonstrated the CW assessment technique based on an ensemble of machine learning and causal reasoning tools. This **technique specifies the new mode** of the NGFR hub for the CW monitoring and prediction [79]. The efficiency of this mode depends on how a smart city is able to provide resources for e-health and cloud computing.−**Guiding intelligent support.** Intelligent assisting tools are a common practice for first responders, e.g., crime prediction [48,49], vital sign detection [15], through-the-wall detectors [16], and stress assistant [24]. In this paper, we emphasize training the NGFR to help people with disabilities. Our systematic reviews were published in papers [38,39,111] as well as in reports [32,33,34,35].−**Collaboration with robot teammates.** First responder teams may also include assistive tools that are autonomous machines such as search-and-rescue robots. Trust in machines is another challenge for the NGFR missions. Monitoring the risk of trust as well as cognitive biases in such a team is of critical issue. We reported the results on the formalization of trust, risk, and bias assessment in such teams in our previous paper [112]. Future first responders will face the challenges of multi-task operation and simultaneous usage of various supporting resources [11,113], building trust in AI-enabled tools by allowing the tracking of risks and biases associated with this trust [112,114,115], as well as Cognitive Workload (CW) monitoring and prediction of the human–robot teams [18,19,20,21], including stress control and prediction [24,25].−**Human digital twins** seem to be the key driver and framework of application AI-enabled tools. This paradigm addresses the area of complex system modeling with essential extension—the active role of humans in the loop. The NGFR hub is a typical perception–action system where monitoring information is learned, accumulated, and represented to the first responder for making decisions. In this paper, CW represents a local perception–action cycle, thus representing a human’s digital twin. We also emphasize the strong relations between the human’s digital twin with other models based on the perception–action cycle, such as self-aware computing and human-in-the-loop processing.

Notably, emergency services for people with disabilities in the context of the NGFR have not been recognized as a separate R&D problem. The solutions were limited to recommendations, e.g., general guidelines for border crossing agencies [31,36], training classes for first responders [36], and recommendations to police and emergency personnel [106]. Our work is pioneering in this area and suggests a systematic approach. The practical value of this approach extends beyond the emergency services, and the concept of the on-body hub is also suited for people with disabilities. This is an essential contribution to the problem of accessibility for people with disabilities, e.g., accessibility to e-health during emergencies via monitoring a person’s vitals using the on-body hub.

## 9. Conclusions

The key conclusion of our study is that the NGFR on-body communication and computing hub is an adequate response to smart city emergency requests and needs, that is, the NGFR hub satisfies the conditions of embedding into smart city technology infrastructure. Our work shows that the NGFR hub has the ability to aggregate previous experience and recent technological innovations. This is the framework for synchronization and synergy R&D for NGFR. Apart from these technical-centric conclusions, the NGFR hub can be considered the technological-societal readiness benchmark of smart cities to delegate advanced opportunities and resources to save people’s lives and properties. This is especially relevant to the potential threats of climate change-induced disasters.

This paper identifies trends and open problems, including technological-societal challenges of the NGFR hub, such as their evaluation and benchmarking, inclusiveness to citizens with disabilities, and the NGFR safety and health monitoring during the mission. We also discuss technology roadmapping using the notion of the EMC. This comprehensive approach positions the proposed concept of the NGFR hub as a practice-centric response from the R&D community to the current and future challenges faced by emergency services in smart cities.

## Figures and Tables

**Figure 1 sensors-24-02366-f001:**
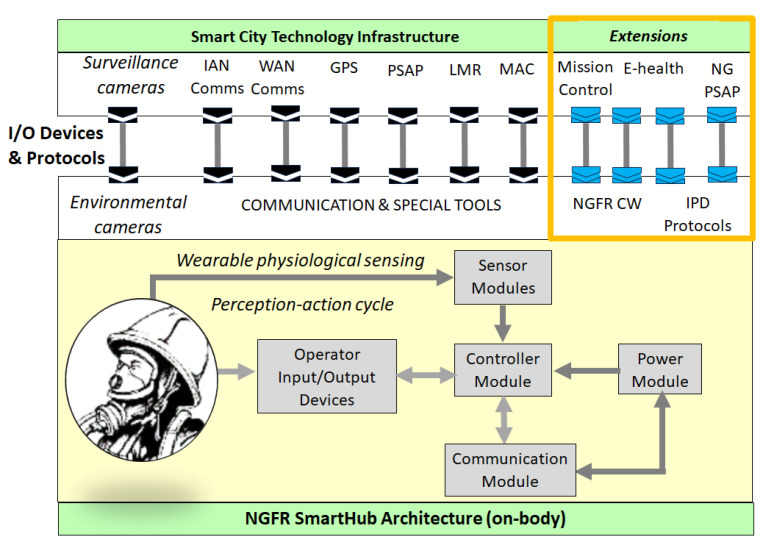
Preparedness for embedding the NGFR SmartHub architecture on-body components with proposed extensions into the smart city technology infrastructure.

**Figure 2 sensors-24-02366-f002:**
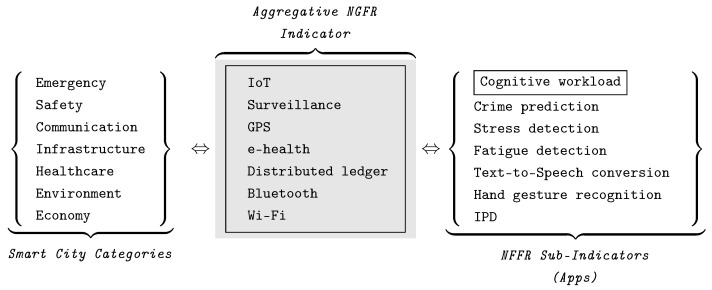
Fragment of embedding procedure: construction of the aggregative NGFR indicator using the smart city categories, essential NGFR indicators, and sub-indicators (apps).

**Figure 3 sensors-24-02366-f003:**
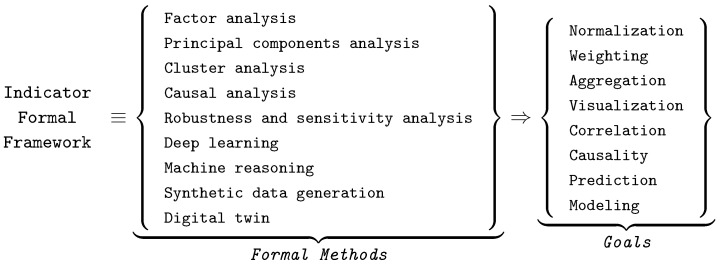
Formal notions and framework of indicator identification and processing.

**Figure 4 sensors-24-02366-f004:**
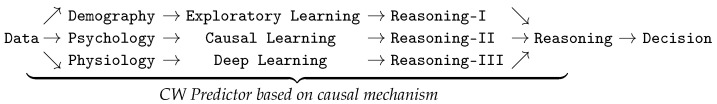
Formal notion of CW mode of the NGFR hub.

**Figure 5 sensors-24-02366-f005:**
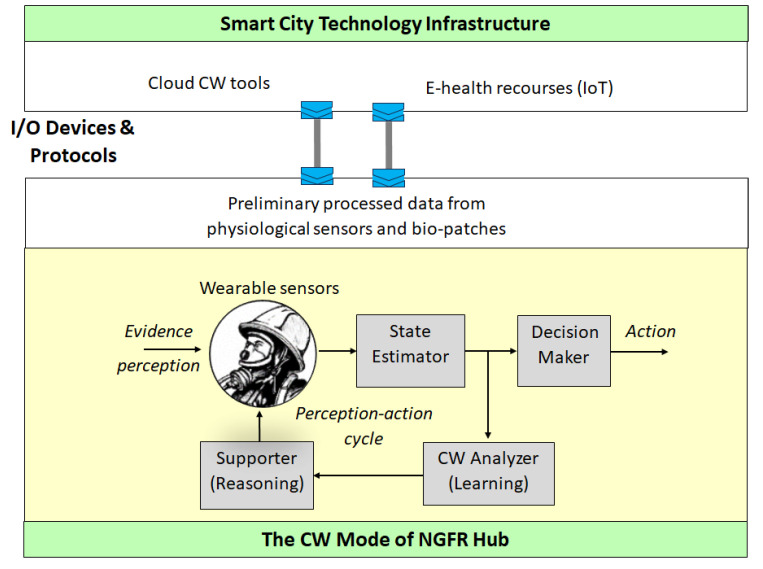
Embedding the CW monitoring mode of NGFR hub.

**Figure 6 sensors-24-02366-f006:**
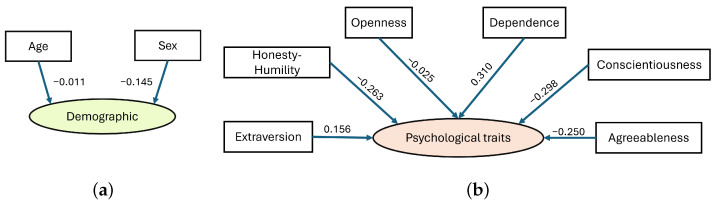
Fragments of exploring demographic traits (**a**) and psychological traits (**b**) using SEM. The regression coefficients are assigned to the arcs between the nodes; the *p*-values are not shown, since for all shown pairs of variables, the Chi-Square test resulted in the *p*-value <0.05.

**Figure 7 sensors-24-02366-f007:**
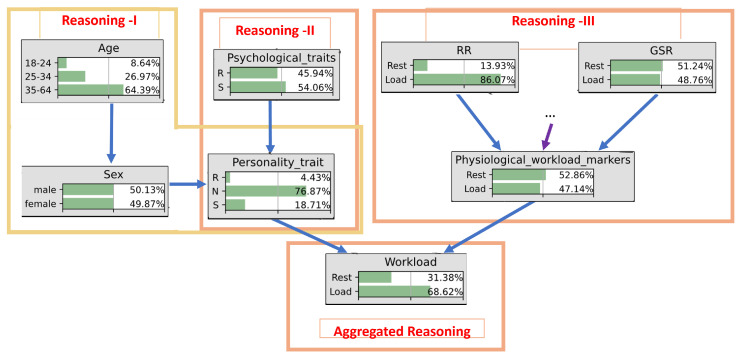
Fragment of the Bayesian causal network that represents the computational scheme of CW in Figure 4.

**Figure 8 sensors-24-02366-f008:**
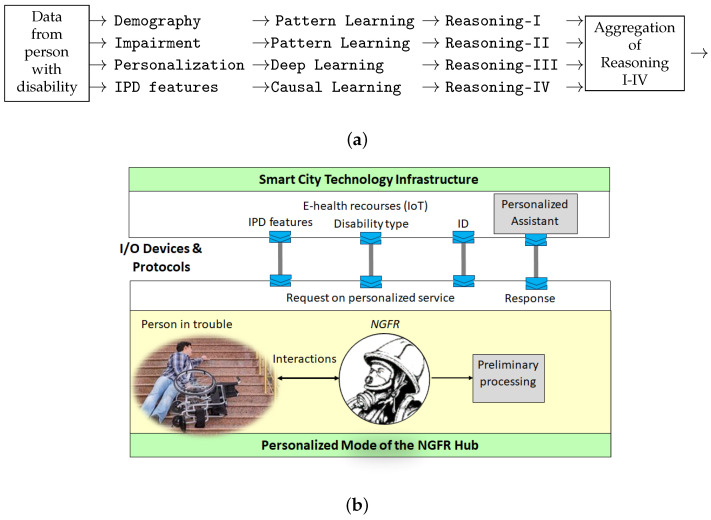
AI formalism of the personalized assistant for people with disabilities (**a**) and its implementation as a personalized mode of the NGFR hub (**b**).

**Figure 9 sensors-24-02366-f009:**
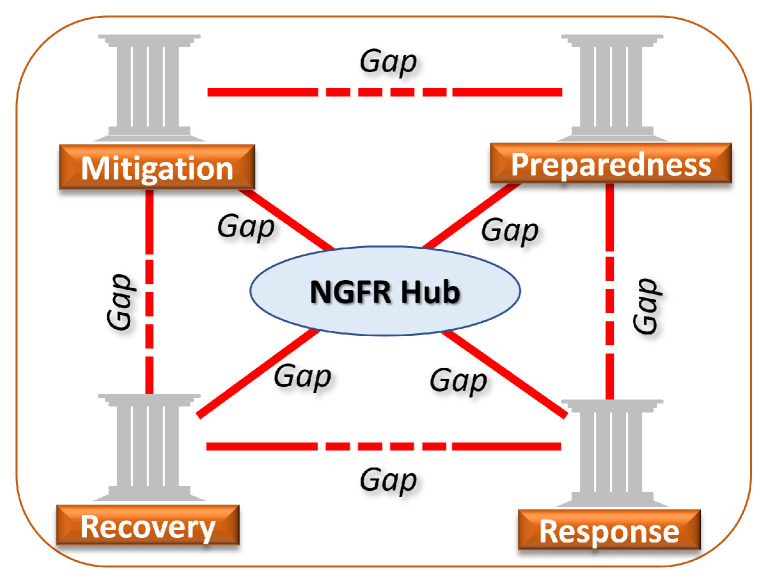
Identification of the technological-societal gaps in R&D of the NGFR hub using the EMC.

**Table 2 sensors-24-02366-t002:** Examples of the CW predictor for NGFR.

Prediction Task	Prediction Context
Prediction strategy: $\underset{\underset{30}{︸}}{\mathrm{Age}}⟶\underset{\underset{\mathrm{Male}}{︸}}{\mathrm{Sex}}⟶\underset{\underset{0\%\;R,\;82\%\;N,\;18\%\;S}{︸}}{\mathrm{Personal}\_\mathrm{trait}}⟶\underset{\underset{\mathrm{Neutral}}{︸}}{\mathrm{Reasoning}-I}$	
How would seeing X change my belief in Y?	Based on preliminary information about the mission, the first responder is a 30-year-old man. The prediction mechanism implemented by the BN path using only demographics results in the Neutral (N) status of this first responder (Reasoning-I).
Prediction strategy: $\underset{\underset{\mathrm{Stressed}}{︸}}{\mathrm{Psychological}\_\mathrm{trait}}⟶\underset{\underset{0\%\;R,\;66\%\;N,\;33\%\;S}{︸}}{\mathrm{Personal}\_\mathrm{trait}}⟶\underset{\underset{\mathrm{Neutral}\;\mathrm{tends}\;\mathrm{to}\;\mathrm{Stressed}}{︸}}{\mathrm{Reasoning}-\mathrm{II}}$	
What if? What if I do *X*?	The prediction mechanism implemented by the BN path using demographic data and psychological traits results in the change in the Neutral (N) status to a Stressed one (S) (Reasoning-II).
Prediction strategy: $\underset{\underset{\mathrm{increasing}}{︸}}{\mathrm{GSR}\mathrm{and}\mathrm{HR}}⟶\underset{\underset{83\%\;\mathrm{CW},\;16\%\;\mathrm{Rest}}{︸}}{\mathrm{Physiological}\_\mathrm{workload}\_\mathrm{markers}}⟶\underset{\underset{\mathrm{CW}}{︸}}{\mathrm{Reasoning}-\mathrm{III}}$	
Why? Was it *X* that caused *Y*? What if I had acted differently?	Assume the availability of the measured GSR data whereas HR data are unavailable. The prediction mechanism implemented by the BN path utilizing only GSR data identify a low CW (Rest) for this first responder. However, upon the resumption of the sensor that provides the HR, along with GSR, the prediction mechanism implies the likelihood of 83% for the high CW (Load) (Reasoning-III).
Prediction strategy: $\begin{array}{ll} \mathrm{Reasoning}-I\;(\mathrm{Neutral}) & \;↘\\ \mathrm{Reasoning}-\mathrm{II}\;(\mathrm{tends}\mathrm{to}\mathrm{Stressed}) & \;\to \\ \mathrm{Reasoning}-\mathrm{III}\;(\mathrm{CW}83\%) & \;↗ \end{array}\underset{\underset{91\%\mathrm{CW},\;9\%\mathrm{Rest}}{︸}}{\mathrm{Reasoning}}⟶\underset{\underset{\mathrm{High}\;\mathrm{CW}\;\mathrm{Risk}}{︸}}{\mathrm{Decision}}$	
Why? Was it X that caused Y?	Reasoning-I indicates a Neutral (N) personality trait of the first responder. Reasoning II suggests a tendency towards a Stressed (S) trait. The BN predicts a high CW (Load) by the HR and GSR sensory data in Reasoning-III, which utilizes deep learning. This reasoning leads to the decision node ’Workload’, indicating a high CW (Load). The real-time updates provide the team leader to respond effectively to the individual team member’s well-being and performance needs, ensuring optimal support and management of the situation.

## Data Availability

The original dataset for the case study is publicly available in GitHub at https://github.com/MartinGjoreski/martingjoreski.github.io/tree/master/files and cited here [83]. The results of the analysis data are contained within the article.

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
