# Peer review of "Next Generation Computing and Communication Hub for First Responders in Smart Cities"

_sensors, 2024, doi:10.3390/s24072366_

Round 1
Reviewer 1 Report
Comments and Suggestions for Authors
This article proposes a new generation first responders concepts to enhance proactiveness of smart cities in crises and disasters. Further it aims to achieve cognitive workload monitoring and provide assistive technologies for individual with disabilities. Although interesting, the reviewer has identified below concerns with the manuscript.
1. Abstract does not clearly highlight the contribution. It seems the concept is described from here and there and very difficult to understand the key objective. Rewriting the abstract is highly recommended.
2. Too many keywords. be specific and use the ones that exactly describes the proposed work.
3. There are certain places throughout the article where text is used as it is in existing article. This problem must be addressed before considering for publication.
4. The connection between human digital twin and machine reasoning model should be discussed clearly.
5. If CW is replicated in the digital human twin, it should be justified through the machine reasoning model isn't it? However this portion is unclear to the reviewer.
6. Reviewer feels like the authors can refer to these articles for "audit of these and other works" section.
"Algorithmic implementation of deep learning layer assignment in edge computing based smart city environment"
"Mobile robotic platform for contactless vital sign monitoring"
"Sensor-based Vital Sign Monitoring, Analysis and Visualisation for Ageing in Place"
Comments on the Quality of English Language
Must be improved. Very lengthy statements which make the reader dizzy when trying to understand.
Author Response
The authors are grateful for the reviews and comments to the
Manuscript ID: sensors-2890720
Title: Next Generation Computing and Communication Hub for First Responders in Smart Cities
Authors: Olha Shaposhnyk*, Kenneth Lai, Gregor Wolbring, Vlad Shmerko, Svetlana Yanushkevich
Received: 19 Feb 2024
Submitted to the special issue
Sensing Technology for Smart Cities: Data, Analytics, and Visualizations
https://www.mdpi.com/journal/sensors/special_issues/26ZFFTU716
Point-by-point response to Comments and Suggestions for Authors:
Comment 1:
Abstract does not clearly highlight the contribution. It seems the concept is described from here and there and very difficult to understand the key objective. Rewriting the abstract is highly recommended.
Author response:
The Authors agree with the reviewer's comment and re-wrote the Abstract (please, see the revised version of the paper, the new abstract text is highlighted using the red font color).
Comment 2:
Too many keywords. be specific and use the ones that exactly describes the proposed work.
Author response:
The Authors agree with the reviewer's comment and reduced the keyword list as follows:
First responders; smart city; computing and communication hub cognitive workload; machine learning and reasoning; persons with disabilities; emergency management cycle.
Comment 3:
There are certain places throughout the article where text is used as it is in existing article. This problem must be addressed before considering for publication.
Author response:
The Authors agree with the reviewer's comment and deleted Section 8.2.
Comment 4:
The connection between human digital twin and machine reasoning model should be discussed clearly.
Author response:
The Authors agree with the reviewer's comment and Subsection 6.4 is now rewritten.
Comment 5:
If CW is replicated in the digital human twin, it should be justified through the machine reasoning model isn't it? However, this portion is unclear to the reviewer.
Author response:
The Authors agree with the reviewer's comment. Subsection 6.4 is renamed and rewritten as follows (lines 449-481).
6.4. The CW mode of NGFR hub
The CW monitoring process is described in the previous subsection as well as schematically illustrated in Fig. 4 (accept feedback). Fig. 5 provides further details and illustrates two structural components:
- The NGFR hub’s CW mode as a complete process of the CW monitoring in terms of the perception-action cycle (lower pane), and
- Embedding the NGFR hub into a smart city infrastructure (upper pane)
These structural components are connected via I/O devices and protocols. The smart city infrastructure provides both cloud-based resources (e.g., machine learning and reasoning) and e-health resources (e.g., personal health information of first responders) to the NGFR hub. Also, the NGFR hub transmits the preliminarily processed data from physiological sensors and bio-patches to further processing on cloud tools.
Consider the operation within the CW mode. The evidence such as the signals from the on-body wearable sensor network, are processed to estimate the current health status of a first responder (State Estimator) and inform the human manager (Decision Maker) who can make decisions, e.g., halt the mission. As the mission continues, the CW analyzer learns this data (CW Analyzer); this learning results in the creation of the CW model composed of both the statistics-based pattern learning and the probabilistic reasoning models such as BN. The BN models the scenario in the form of a joint probability distribution. It is constructed using joint probability distribution, derived from pattern recognition as well as other statistical information. The BN is the framework for reasoning resulting in the risk assessment (Supporter). For details, we refer the reader to a wide spectrum of literature, including the recent monograph \cite{[Murphy-2022]}. The assessed risks are reported to the first responder, e.g., ``The CW increased by 20% and reached level 3 (medium on the cale of 1-5) in the past time interval (5 min)’’. In the next time interval, the health status of the first responder is changed, causing the behaviour change in the adversarial environment (flames, falling debris), and the BN model is updated. This computational model of CW is a digital replica of a physical object’s CW status. It continuously evolves in near real-time following the behaviour of this object \cite{ [Miller-2022],[Thelen-2022-Part_1],[Thelen-2022-Part_2]}.
Note that Fig. 5 reflects several advanced computational paradigms for the implementation scheme. First, it is a self-aware computing \cite{[Bauer-2020],[Hoffmann-2020]}. The applications of the self-awareness principle for CW tasks are reported, for example, in \cite{[Dell-Agnola-2021],[Masinelli-2020]}, where a 78% accuracy of CW detection was achieved. Secondly, this is a human-in-the-loop paradigm.
The authors added a new reference to explain the text above.
[Murphy-2022]
P. Murphy, Probabilistic Machine Learning. An Introduction. The MIT Press, Cambridge, Massachusetts, London, England, 2022.
Comment 6:
Reviewer feels like the authors can refer to these articles for "audit of these and other works" section
"Algorithmic implementation of deep learning layer assignment in edge computing based smart city environment",
"Mobile robotic platform for contactless vital sign monitoring",
"Sensor-based Vital Sign Monitoring, Analysis and Visualisation for Ageing in Place"
Author response:
The Authors agree with the reviewer's comment and referred to the mentioned papers:
[Kerr-2018]
Sensor-based Vital Sign Monitoring, Analysis and Visualisation for Ageing in Place,
Kerr, E. P. ; Coleman, S.A. ; Kerr, D. ; Vance, P. ; Gardiner, B. ; Zhang, Y. ; Wang, F. ; Wu, C. 2018
International Joint Conference on Neural Networks (IJCNN), 2018, p.1-7
[Lee-2021]
Algorithmic implementation of deep learning layer assignment in edge computing based smart city environment, Lee, Kyuchang ; Silva, Bhagya Nathali ; Han, Kijun
Computers \& electrical engineering, 2021-01, Vol.89, p.106909, Article 106909
[Huang-2022]
Mobile Robotic Platform for Contactless Vital Sign Monitoring
Huang, Hen-Wei ; Chen, Jack ; Chai, Peter R ; Ehmke, Claas ; Rupp, Philipp ; Dadabhoy, Farah Z ; Feng, Annie ; Li, Canchen ; Thomas, Akhil J ; da Silva, Marco ; Boyer, Edward W ; Traverso, Giovanni
Cyborg and bionic systems, 2022-01, Vol.2022
Best regards,
AUTHORS: Olha Shaposhnyk , Kenneth Lai, Gregor Wolbring , Vlad Shmerko, and Svetlana Yanushkevich.

Reviewer 2 Report
Comments and Suggestions for Authors
The paper is timely and well written. Some modifications would help in making the context clearer.
Safety monitoring aimed to hurban areas using WSNs is a wide adopted scenario [a] . A short elaboration on the integration of interactions between NGFR and the aforementioned systems can be introduced. [a] https://doi.org/10.1145/3627345.3627359
In the abstract, line 9: "we demonstrate how the NGFR 9 hub can be employed for cognitive workload monitoring of first Second," is not clear.
The authors can briefly summarize the main findings and discussions of this paper in the introduction to make the context more clear.
Check figure 9.
Author Response
The authors are grateful for the reviews and comments to the
Manuscript ID: sensors-2890720
Title: Next Generation Computing and Communication Hub for First Responders in Smart Cities
Authors: Olha Shaposhnyk*, Kenneth Lai, Gregor Wolbring, Vlad Shmerko, Svetlana Yanushkevich
Received: 19 Feb 2024
Submitted to the special issue
Sensing Technology for Smart Cities: Data, Analytics, and Visualizations
https://www.mdpi.com/journal/sensors/special_issues/26ZFFTU716
Point-by-point response to Comments and Suggestions for Authors:
Comment 1:
Safety monitoring aimed to urban areas using WSNs is a wide adopted scenario [a] . A short elaboration on the integration of interactions between NGFR and the aforementioned systems can be introduced. [a] https://doi.org/10.1145/3627345.3627359
Author response:
The authors agree with these remarks, and the work is now referred to in the References.
[Ragnoli-2022]
Ragnoli, Mattia ; Leoni, Alfiero ; Barile, Gianluca ; Ferri, Giuseppe ; Stornelli, Vincenzo,
LoRa-Based Wireless Sensors Network for Rockfall and Landslide Monitoring: A Case Study in Pantelleria Island with Portable LoRaWAN Access, Journal of low power electronics and applications, 2022-09, Vol.12 (3), p.47
Comment 2:
In the abstract, line 9: "we demonstrate how the NGFR 9 hub can be employed for cognitive workload monitoring of first Second," is not clear.
Author response:
The Authors agree with the reviewer's comment and re-wrote the Abstract (please, see the revised version of the paper, the new abstract text is highlighted using the red font color).
Comment 3:
The authors can briefly summarize the main findings and discussions of this paper in the introduction to make the context more clear.
Author response:
The Authors agree with the reviewer's comment.
The following summary is added (lines 82-90 of the original version, now lines 85-104):
In summary, the key motivation of this work is to examine and extend the concept of SmartHub, a breakthrough practice-centric solution to support the new generation of first responders (NGFR). The NGFR must be enabled to execute multiple tasks under incomplete and uncertain information, and unreliable equipment and external support in the adversarial environment. We outline the R&D roadmapping and provide strategic direction to consolidate the R&D efforts.
Our main goal is to lay out the ways to embed the SmartHub into the smart city. We outline how to satisfy the requirements for such embedding, in terms of standard categories and indicators of smart city performance. We focus on conditions of critical importance such as interoperability and standardization, the potential for extensions, modeling, and improving NGFR technologies to assist citizens with disabilities.
Our extensions of the SmartHub address two complex tasks: CW monitoring and emergency assistance for people with disabilities. In solving complex problems, the aggregation of different methods often provides reliable assessments of events and processes involved in such. Following this prerogative, we developed an AI formalism - an ensemble of machine learning processes aggregated using machine reasoning for both tasks.
These multiple targets prompt the second goal of our study, -- synchronization of the R&D towards the smart city’s resilience to crises and disasters. This strategic goal leads to the commonly accepted Emergency Management Cycle (EMC) doctrine that is innovatively interpreted in our work in terms of the R&D technology gaps.
The quintessence of our approach is the NGFR hub, the modified version of the SmartHub.
Comment 4:
Check figure 9
Author response:
The Authors checked the content and deleted the subsection and this figure to make the text more concise.
Best regards,
AUTHORS: Olha Shaposhnyk , Kenneth Lai, Gregor Wolbring , Vlad Shmerko, and Svetlana Yanushkevich.

Reviewer 3 Report
Comments and Suggestions for Authors
The Abstract must be improved following the requirements: Goal, Method, Results, and Originality.
The question "– Does the proposed NGFR hub ’cover’ the technology infrastructure of a smart city?" from the Introduction section is not a Problem of research. In correspondence with the content of section 3, it is not clear what the research's objective or goal is.
Sections 4 and 5 are formulated in a non-scientific manner. The formulations are very simple ("Details can be found in [27].") and without sources (see Figures 1, 2, and 3!).
The methodology of research is not clear mentioned.
I recommend organizing the research more clearly, starting from the objective, research question, research hypothesis, research methodology, data and results, discussion, and conclusion. The literature review must be a very important section in the new version. Please avoid the presentation of theory in an excessive way! There is a "problem formulation" in section 7.
Finally, it is recommended that the paper be reorganized with a clear objective and a clear methodology.
For all of these reasons, I recommend a major revision.
Author Response
The authors are grateful for the reviews and comments to the
Manuscript ID: sensors-2890720
Title: Next Generation Computing and Communication Hub for First Responders in Smart Cities
Authors: Olha Shaposhnyk*, Kenneth Lai, Gregor Wolbring, Vlad Shmerko, Svetlana Yanushkevich
Received: 19 Feb 2024
Submitted to the special issue
Sensing Technology for Smart Cities: Data, Analytics, and Visualizations
https://www.mdpi.com/journal/sensors/special_issues/26ZFFTU716
Point-by-point response to Comments and Suggestions for Authors:
Comment 1:
The Abstract must be improved following the requirements: Goal, Method, Results, and Originality.
Author response:
The Authors agree with the reviewer's comment and re-wrote the Abstract (please, see the revised version of the paper, the new abstract text is highlighted using the red font color).
Comment 2:
The question "– Does the proposed NGFR hub ’cover’ the technology infrastructure of a smart city?" from the Introduction section is not a Problem of research. In correspondence with the content of section 3, it is not clear what the research's objective or goal is.
Author response:
The Authors agree with the reviewer's comment and deleted this sentence; the authors added this material in Section 3 in the form of a research question.
Comment 3:
Sections 4 and 5 are formulated in a non-scientific manner. The formulations are very simple ("Details can be found in [27].") and without sources (see Figures 1, 2, and 3!).
Author response:
The authors partially agree with this comment and did their best to improve the text. In particular:
- The authors do not agree with the comment on a ``non-scientific manner’’ in sections 4 and 5, that is,
- a) not employed in science,
- b) not conforming to the principles or methods of science, and
- c) not demonstrating scientific knowledge or scientific methods.
On the contrary, sections 4 and 5 reflect a reported trend and a breakthrough solution in the area of NGFR, i.e. on-body communication and hub, and edge composite technologies. The references in this paper clearly indicate that the on-body hub manifests a synergy of scientific achievements.
- The proposed results could be viewed as a logical extension of paper [27] (Serrano et al., NIST Special Publication 2022); the significant extensions include:
- a) an application to the NGFR technology (this topic is absent in [27]);
- b) a relationship between the NGFR hub and assistive technologies for people with disabilities (this topic is absent in [27]);
- c) the alternative categories and indicators [59-60] (this topic is absent in [27]);
- d) the fundamentals of cognitive workload assessment for the NGFR and people with disabilities (this topic is absent in [27]);
- e) a mandatory part of smart city related to projections of the Emergency Management Cycle (EMC) (this topic is absent in [27]);
- f) a practice of interoperability in the smart city (this topic is absent in [27]);
- g) a technology roadmapping toward potential threats (this topic is absent in [27]); and
- h) benchmarking to satisfy the requirements of predictive modeling (this topic is absent in [27]).
- Figures 1, 2 and 3 are authors original, and not sourced from the internet.
Comment 4:
The methodology of research is not clear mentioned.
Author response:
The authors agree with this comment.
Indeed, the authors did not use the term ``methodology’’. Instead, the authors use a broader term ``approach’ as related to the context of the paper. Our approach and its details are introduced throughout the paper, for example, in the problem formulation (section 3). However, to avoid potential misunderstandings by the readers, the authors include the notion of methodology in the revised version. We make minor corrections in section 3 to emphasize the used methods and methodologies:
- Auditing methods for the 1st task,
- embedding methods for the 2nd task,
- experimental sondage (exploration) methodology for the 3rd task,
- adaptation methodology for the 4th task, and
- discussion methods for the 5th
Comment 5:
I recommend organizing the research more clearly, starting from the objective, research question, research hypothesis, research methodology, data and results, discussion, and conclusion. The literature review must be a very important section in the new version. Please avoid the presentation of theory in an excessive way! There is a "problem formulation" in section 7.
Author response:
The Authors agree with the reviewer's comment and implemented the following corrections to satisfy the recommendations of the reviewer:
- The Abstract is rewritten.
- The Literature review section was partially updated.
- The ``objective, research question, research hypothesis, research methodology’’ are corrected in the revised abstract and in the revised section 3 (please, see the revised version of the paper, the new text is highlighted using the red font color).
- The remark `` to avoid the presentation of theory in an excessive way’’ was addressed by partial corrections of the text. Specifically, the authors restructured sections 3, and partially 4 and 5 (please, see the revised version of the paper, the new text is highlighted using the red font color).
- The problem formulation in section 7 addresses the content of section 7 only. However, to avoid misunderstanding, we replaced the subsection 7.1 title with ``NGFR interactions with individuals with disabilities’’.
Comment 6:
Finally, it is recommended that the paper be reorganized with a clear objective and a clear methodology
Author response:
The Authors agree with the reviewer's comment and implemented the corresponding corrections, as well as made section 3 - Problem formulation and approach - clearer. The notion of methods and methodology were included in section 3 as indicated in the answers to Comment 4.
Comment 7:
For all of these reasons, I recommend a major revision.
Author response:
The authors implemented the reviewers’ suggestions and corrections as required by this and other reviewers.
Best regards,
AUTHORS: Olha Shaposhnyk , Kenneth Lai, Gregor Wolbring , Vlad Shmerko, and Svetlana Yanushkevich.

Round 2
Reviewer 1 Report
Comments and Suggestions for Authors
Authors have successfully addressed all my major concerns. However, I think language editing should be done before considering for publication as some sections are still difficult to understand at one glance.
Comments on the Quality of English LanguageMinor editing required to improve the quality and understandability
Author Response
The authors thank the reviewer for the comments.
The authors will conduct the thorough language editing as recommended by the reviewer.
Reviewer 3 Report
Comments and Suggestions for Authors
After the evaluation of the changes and a reanalysis of the content of this article, in complete agreement with the research question assumed by the authors, taking into account the answers and adjustments made by the authors, I consider that the material presented does not make a contribution in the field proposed for research. Moreover, the perspective in which the results are presented is of a low scientific degree.
In this context, I maintain the opinion formulated in the previous review. Still, I propose rejecting the paper until a serious redesign of the entire paper is considered.
Author Response
The authors thank the reviewer for the review of the revised paper. However, the authors do not agree with the new comments by Reviewer 3 for the following reasons:
- The authors made every effort to respond to round 1 comments by Reviewer 3 and made corrections.
- Reviewer 3 provided strong criticism of the fundamentals of the authors’ work, in particular, on the topic, scientific contribution, and practical value. This is beyond any common reviewer practice.
- The authors state that this paper
- Pioneers in bridging the essential gap existing in assistive technologies for people with disabilities.
- Represents the novel practice-centric extensions of the first responder technologies, and proves it through the case study experiments.
- Satisfies the standard requirements of this journal and this special issue.